# Protective Effect of HER2 Gene Polymorphism rs24537331 in the Outcome of Canine Mammary Tumors

**DOI:** 10.3390/ani13081384

**Published:** 2023-04-18

**Authors:** Ana Canadas-Sousa, Marta Santos, Patrícia Dias-Pereira

**Affiliations:** 1Department of Pathology and Molecular Immunology, Instituto de Ciências Biomédicas Abel Salazar, ICBAS-UP, University of Porto, 4050-313 Porto, Portugal; 2Department of Microscopy, Instituto de Ciências Biomédicas Abel Salazar, ICBAS-UP, University of Porto, 4050-313 Porto, Portugal

**Keywords:** canine mammary tumor, HER2 gene, genetic variance, SNP, survival

## Abstract

**Simple Summary:**

Human epidermal growth factor receptor 2 (HER2) participates in breast cancer pathogenesis and progression. However, the role of HER2 in canine mammary tumors is not completely understood, and data from different immunohistochemical studies are conflicting. A remarkable genetic variation of the canine HER2 gene has been recognized, including several single nucleotide polymorphisms (SNPs), which may account for the contradictory results. This study aims to assess the relationship between SNPs rs24537329 and rs24537331 in canine HER2 gene and clinicopathological features, clinical progression, and outcome of canine mammary tumors (CMT). Our results demonstrated that SNP rs24537331 was associated with decreased tumoral necrosis and with longer disease-specific overall survival; however, no significant associations were found between SNP rs24537329 and the tumors’ clinicopathological characteristics or survival. Our data suggest that SNP rs24537331 may have a protective effect in CMT, allowing the identification of a subgroup of animals prone to develop less aggressive forms of the disease. This investigation emphasizes the importance of the animal’s genetic background assessment as a valuable tool to be used by clinicians and oncologists in CMT management.

**Abstract:**

The role of HER2 in canine mammary tumors is not completely elucidated, and the contradictory results published so far may be, in part, explained by the genetic variability recognized in the canine HER2 gene. Single nucleotide polymorphism (SNPs) in HER2 were recently associated with less aggressive canine mammary tumor histotypes. This study assesses the relationship between SNPs rs24537329 and rs24537331 in canine HER2 gene and clinicopathological characteristics and outcome of mammary tumors in a group of 206 female dogs. Allelic variants were observed in 69.8% and 52.7% of the dogs for SNP rs24537329 and rs24537331, respectively. Our results demonstrated that SNP rs24537331 was associated with decreased tumoral necrosis (HR: 3.09; *p* = 0.012) and with longer disease-specific overall survival (HR: 2.59; *p* = 0.013). However, no statistically significant associations were found between SNP rs24537329 and the tumors’ clinicopathological characteristics or survival. Our data suggest that SNP rs24537331 may have a protective effect in canine mammary tumors, allowing the identification of a subgroup of animals prone to develop less aggressive forms of the disease. This study emphasizes the importance of the genetic tests associated with clinical images and histological examinations when assessing CMT outcomes.

## 1. Introduction

Recent advances in molecular, genetic, and clinical research contributed to a noticeable development in understanding cancer biology fundamentals. Cancer is recognized as a complex, multifactorial, and multistep disease encompassing a myriad of genetic, epigenetic, and phenotypic changes. The disease is characterized by abnormal gene structure or function that are responsible for the conversion of normal cells into neoplastic ones. The development of new molecular tools allowed a better characterization of human breast cancer, emphasizing its genetic background. Indeed, comprehensive research in molecular genetics applied to oncology led to the identification of genetic profiles associated with clinicopathological features, progression, and response to treatment of breast cancer [1,2].

Canine mammary tumors (CMT) represent the most frequently diagnosed tumors in female dogs, depicting 30% to 70% of all neoplasms in this animal species. The incidence of CMT increases with the animal age. Mammary tumors are rare in very young dogs, and most cases occur in middle-aged to old animals, with a peak incidence observed at 11 to 13 years of age. The hormonal background plays a key role in CMT development. In fact, almost all cases occur in females, and the rare mammary tumors described in males are associated with estrogen-producing testicular neoplasms. Moreover, ovariectomy carried out in young animals has a protective effect against CMT development, while long-term progestogens administration is associated with an increased CMT risk. These reproductive health policies may explain the considerable differences observed in the incidence and prevalence of CMT worldwide. The histological picture of CMT is characterized by a remarkable heterogeneity and may include different cell populations of luminal epithelial, myoepithelial, or mesenchymal origin, resulting in the recognition of several histological types and subtypes. In this way, CMT comprises a heterogeneous group of diseases with different clinicopathological features and biological behavior, thus representing a significant prognostic challenge [3,4,5,6,7]. The identification of features that can serve as an auxiliary tool in the management of CMT is, therefore, crucial. The great heterogeneity observed in the morphology and clinical evolution of CMT can be partly attributed to the genetic variability of the hosts. In fact, over the last few years, several studies demonstrated that the animal’s genetic profile influences not only the risk of developing CMT but also various clinicopathological characteristics and the progression of the disease [8,9,10,11,12].

HER2 (human epidermal growth factor receptor 2) is a tyrosine kinase receptor belonging to the epidermal growth factor receptor family that plays an important role in breast cancer pathogenesis and progression, participating in the regulation of cell proliferation, differentiation, and migration, as well as in angiogenesis and cell survival [13]. It constitutes a crucial biomarker in the molecular classification of breast cancer, routinely used as a prognostic and predictive factor. HER2 overexpression is found in around 20 to 30% of breast cancers, representing an aggressive subtype associated with adverse clinicopathological features and poor prognosis, namely high recurrence rates, short disease-free survival, and short overall survival [14,15]. HER2 status is mandatory to determine the suitability of breast cancer to HER2 targeted therapy [15].

The role of HER2 in CMT is less understood. Although the immunohistochemical expression of this receptor in CMT has been widely investigated, the results of different studies are conflicting; some authors have not observed HER2 immunoreactivity [16,17], albeit others described variable degrees of immunostaining in mammary tumors in this species [18,19,20,21]. Furthermore, a consensus has not yet been established regarding its value as a prognostic factor. While some authors associated HER2 overexpression with aggressive clinicopathological features, namely high mitotic index, high histological grade, and marked nuclear pleomorphism [18,19,22], others failed to demonstrate such associations [20]. In other studies, HER2 overexpression was even related to a prolonged survival time [21,22].

These discrepant results on the role of HER2 overexpression in CMT can be due to variations in the immunohistochemical protocols (using different primary antibodies, concentrations, or incubation times) and evaluation systems employed in the different studies [23] or can be, in part, related to variations in the HER2 gene. Although the presence of the receptor in mammary tissues has been confirmed through immunohistochemical analyses, its functional activity may be altered due to genetic variation. Some authors have recognized that genetic variability of the canine HER2 gene exists and includes amplifications, translocations, and genetic single nucleotide polymorphisms (SNP) [20,21,24,25]. Although our group did not find an association between SNPs in the canine HER2 gene and the risk for CMT development [26], more recently, the influence of HER2 SNPs on histotype aggressiveness of CMT was demonstrated [12].

To elucidate the association between HER2 and biology and prognosis of CMT, in this study, we aim to assess the association between SNPs rs24537329 and rs24537331 in canine HER2 gene and clinicopathological features, clinical progression, and outcome of CMT.

## 2. Materials and Methods

A study was conducted including 206 female dogs with histologically confirmed mammary tumors. Dogs were treated with radical unilateral mastectomy (removal of an entire unilateral mammary chain) and/or partial mastectomy (removal of one to three mammary gland pairs, including the affected ones). Owners provided consent for surgery with curative intents, as well as for the use of the material for research purposes. This protocol was approved by the Ethics Committee of the Institute of Biomedical Sciences Abel Salazar, University of Porto (ORBEA; P151/2016).

Mammary specimens were collected after surgery, fixed immediately in a 10% buffered formalin solution, and routinely processed for histological examination. For each case, the age at the time of the diagnosis, the number of tumors, and the tumor size—corresponding to the largest diameter measured by the same pathologist (ACS) during trimming—were recorded. The histological diagnosis was established by a consensus of three pathologists (ACS, MS, and PDP) under a multi-head microscope, according to the criteria of the most recent official classification [27]. The tumoral growth pattern was evaluated and classified as expansive when it was delimited with a capsule, or infiltrative when not delimited by a capsule but without signs of vascular invasion or lymph node metastasis, and invasive if vascular invasion or lymph node metastases were observed. In cases of dogs with multiple malignant tumors, a reference lesion was assigned for the statistical analyses, following previously reported criteria. The reference lesion was considered as the tumor presenting peritumoral vascular invasion (primary criterion), higher nuclear pleomorphism (secondary criterion), or the one with the largest diameter (tertiary criterion) [28]. Histological grading was performed by consensus of three pathologists (ACS, MS, and PDP) according to the Nottingham histologic grading method (NHG) [29], based on the assessment of three parameters: tubule formation, nuclear pleomorphism, and mitotic counts. Each parameter of the NHG was scored from 1 to 3, and the total combined score defined the grade, as follows. Tubule formation was scored as 1, 2, or 3 when more than 75%, 10 to 75%, or <10% of neoplastic cells, respectively, were arranged in structures exhibiting an obvious lumen. Nuclear pleomorphism was scored 1 when a slight increase in variability of nuclear size and shape, compared with normal surrounding epithelial cells, was observed; score 2 denoted moderate variation in nuclear size and shape; score 3 displayed marked variation in nuclear size and shape, with very large and bizarre forms. Mitotic figures were counted in 10 high-power fields in the mitotically most active parts of the tumor, avoiding areas with necrosis and large amounts of stroma. The cutoffs for establishing the score of the mitotic counts of the NHG were adjusted to the high-power field (HPF) of the microscope (Nikon E600 Eclipse with a field diameter of 0.55 mm, HPF area of 0.237 mm^2^; Nikon, Tokyo, Japan): score 1, ≤8 mitotic figures; score 2, 9–17 mitotic figures; score 3, ≥18 mitotic figures. The sum of the scores (of 1 to 3 points) obtained in each of the parameters defined the histological grade of the tumor, as follows. A total score ≤5 points, 6–7 points, or 8–9 points denotes grades I, II, and III carcinomas, respectively. A veterinary adaptation of the human Nottingham Prognostic Index (vet-NPI) was also computed [30]. Veterinary-adapted NPI was computed as vet-NPI = [tumor size (cm) × 0.2] + NHG (1, 2, or 3 respectively for grades I, II, and III) + evidence of vascular invasion/regional lymph node metastases (1 or 2 if absent or present, respectively) [25].

Two-year follow-up data were obtained by consulting the medical records and the referring veterinarian. Disease-specific overall survival was calculated from the time of diagnosis to the date of the animal’s death/euthanasia due to the progression of the neoplastic disease. Animals that died or were euthanized for unrelated causes and those that were lost to follow-up were censored, respectively, at the time of death and at the data of their last clinical examination. Euthanasia was performed only in the terminal stage of the disease. A Necropsy examination was performed upon the owner’s consent.

Genomic DNA was extracted from peripheral blood samples (obtained with standard venipuncture) using a High Pure PCR Template preparation kit (Roche, Mannheim, Germany). The DNA quality was evaluated by measuring the optical density, and the quantity was assessed employing the NanoDrop 1000 Spectrophotometer (Thermo Fisher Scientific, Waltham, MA, USA). SNP genotyping was performed using MassARRAY iPLEX Gold Technology at the Unidade de Genómica/Serviço de Genotipagem do Instituto Gulbenkian de Ciência. This technology for SNP genotyping consists of an initial PCR reaction, followed by multiplexed primer extension (using mass-modified dideoxynucleotide terminators of an oligonucleotide primer), which anneals immediately upstream of the polymorphic site of interest. Additionally, the MALDI-TOF (matrix-assisted laser desorption/ionization-time of flight) mass spectrometry allows the recognition of the SNP allele by the different mass of the extended primer [31,32].

Two canine HER2 (dog chromosome 9) synonymous SNPs were studied: rs24537329, located at exon 13, corresponding to a T/C change (Glu-Glu)—wild-type: TT; variant genotype: C allele carriers; and rs24537331, located at exon 14, corresponding to a G/A change (Cys-Cys)—wild-type: GG; variant genotype: A allele carriers.

Statistical analysis of data was performed using the computer software SPSS for Windows (version 26). Chi-square analysis (or Fisher’s exact test, when appropriate) was used to evaluate the significance of the relationship between HER2 SNP and the categorical variables. To assess the differences between malignant and benign tumor age groups, Student’s *t*-test was used for continuous variables. Disease-specific survival curve was computed using the Kaplan–Meier product-limit estimates method and log-rank (Mantel–Cox) test. Cox proportional hazard regression univariable analyses were used to evaluate the prognostic value of the SNPs and other clinicopathological variables and to estimate the risk value using the 95% confidence interval. The survival analyses were performed in cases with at least one malignant tumor and for which follow-up data were available. A 5% level was considered to define statistical significance.

## 3. Results

Of the 206 female dogs enrolled in the study, SNPs amplification succeeded in 205. For the population included in this study, the frequency of SNP rs24537331 was 47.3% for GG, 38.5% for GA, and 14.1% for AA, and of SNP rs24537329 was 30.2% for TT, 42.0% for TC, and 27.8% for CC (Table 1).

The mean age of the enrolled population was 10.15 years old (7–18 years old); 9.68 years for dogs with benign tumors (n = 73/205; 35.61%) and 10.42 years for dogs with at least one malignant tumor (n = 132/205; 64.40%), a statistically significant difference (*p* = 0.028). Multiple synchronous tumors were diagnosed in 65.50% of the cases. Nearly half of the animals included in this study were mixed-breed dogs (n = 93/205; 45.4%), and the others were distributed among several different breeds, the most represented being Poodle (n = 20/205; 9.8%), Cocker Spaniel (n = 12/205; 5.9%), German Shepard (n = 11/205; 5.4%), Boxer (n = 11/205; 5.4%), and Labrador Retriever (n = 9/205; 4.4%). The most common benign tumors were mixed tumors (n = 33), complex adenomas (n = 25), and simple adenomas (n = 11). The most frequent malignant neoplasms were complex carcinoma (n = 21), tubulopapillary carcinomas (n = 17), solid carcinomas (n = 12), comedocarcinomas (n = 10), and mixed carcinomas (n = 10). According to the Nottingham histological grading method, 35 (33.3%), 47 (44.8%), and 23 (21.9%) out of 105 carcinomas, were classified as grade I, II, and III carcinomas, respectively. The histological grade was not assigned for sarcomas and carcinosarcomas. Vascular invasion and lymph node metastases were identified in 16.9% (n = 22) and 26.7% (n = 31) of the cases, respectively.

Two-year follow-up data were available for 115 dogs with malignant tumors. Of those, 57/115 (49.57%) were alive at the end of the follow-up period, while 31/115 (26.96%) died due to the progression of the disease. Animals lost to follow-up and animals that died from causes not related to the mammary neoplasia (n = 27/115; 23.48%) were censored.

Table 2 displays the association between SNPs rs24537331 (wild-type GG vs. A allele carriers) and rs24537329 (wild-type TT vs. C allele carriers) and clinicopathological parameters, namely age at the time of the tumor diagnosis, breed (purebred/mongrel), tumor number and size, and histological classification (benign/malignant); for malignant mammary tumors, pattern of tumor growth, NHG histological grade, NHG histological grading parameters (tubule formation, nuclear pleomorphism, and mitotic index), presence of tumor necrosis, vet-NPI, vascular invasion, and lymph node metastases were also assessed. No significant associations could be established between the studied SNPs and clinicopathological features, except for SNP rs24537331 and tumoral necrosis. Variant allele carriers for SNP rs24537331 had a lower probability of presenting tumoral necrosis when compared to tumors of dogs with wild-type allele (HR: 3.09; CI: 1.26–7.59; *p* = 0.012).

The Kaplan–Meier survival curve and log-rank test (Figure 1) demonstrated that carriers of the variant allele for SNP rs24537331 had a mean disease-specific overall survival of 21.7 months, while the wild-type population had a mean disease-specific overall survival of 18.3 months, a statistically significant difference (*p* = 0.009). This result was further confirmed by Cox univariable regression analysis, which demonstrated that the risk of death due to the neoplastic disease was almost 3 times higher for wild-type animals than for variant allele carriers (HR = 2.59, 95% CI 1.22–5.49, *p* = 0.013). The SNP rs24537329 was not significantly associated with survival.

## 4. Discussion

Technological developments in recent decades have contributed to a comprehensive knowledge of oncology, recognizing cancer as a genetically based disease. During neoplastic transformation, normal cells accumulate a series of genetic alterations that drive functional and phenotypical changes rendering them more resistant, with an unlimited capacity to proliferate, and more aggressive. On the other hand, several investigations have underlined the importance of the individual genetic profile in the definition of several clinicopathological features of cancer, as well as in the biological behavior and prognosis of the disease [1,2]. Indeed, genetic signatures are an effective tool assisting in the implementation of prevention, screening, and early detection protocols in several cancers, including human breast cancer. They also play a relevant role in helping to elect the most appropriate treatment strategies, as well as in predicting individual responses to treatment and clinical outcomes of the neoplastic disease [33].

The HER2 oncogene encodes a transmembrane tyrosine kinase included in the epidermal growth factor receptor family that participates in cell proliferation, differentiation, migration, and survival. HER2 status is a reliable prognostic and predictive factor in breast cancer that influences the progression, biological behavior, and response to treatment [13,34,35,36]. The relevance of HER2 expression in CMT is not straightforward, with different studies sometimes reporting contradictory results. A growing body of evidence has demonstrated a remarkable genetic variation of the canine HER2 gene, with several SNPs being documented [20,21,24,25]. Nonetheless, although SNPs in the human HER2 gene have been associated with the susceptibility, clinical course, and response to treatment of breast cancer disease [37,38,39], the knowledge regarding the influence of HER2 SNPs on clinicopathological features and prognosis of CMT is still incomplete. In previous studies, our group demonstrated that SNP rs24537329 and rs24537331 in the canine HER2 gene were related to the CMT histotype, with the former being associated with the development of aggressive histological types, while the latter was related to less aggressive CMT histotypes [12]. Nevertheless, none of these SNPs were associated with the risk of developing the disease [26].

In the present study, we assessed the relationship between SNPs rs24537329 and rs24537331 in the canine HER2 gene and clinicopathological characteristics and outcomes of CMT.

Our results demonstrated that genetic variants for SNP rs24537331 were less likely to display tumoral necrosis; indeed, only 34% of the malignant tumors of variant allele carriers displayed this histological feature, while almost 66% of the wild-type animals presented a malignant tumor with necrosis. Necrosis is a frequent histological characteristic of several solid tumors, including mammary neoplasms [40]. Tumoral necrosis involves hypoxia, the release of proinflammatory factors, the development of an inflammatory response, and angiogenesis, processes that support tumor growth and contribute to a poor prognosis [41]. In fact, tumoral necrosis has been associated with other aggressive clinicopathological features of CMT, namely high proliferative activity, reduced E-cadherin expression, and increased FoxP3 regulatory T cells, among others [42,43,44]. Additionally, necrosis is a critical feature in establishing the diagnosis of a specific subtype of CMT, comedocarcinoma, which is associated with unfavorable outcomes [28,45]. Additionally, the present data showed that SNP rs24537331 variant allele genotypes were associated with longer disease-specific overall survival. Wild-type animals had a nearly three times higher risk of death due to the neoplastic disease during the follow-up period than carriers of the variant allele.

No statistically significant associations were found between SNP rs24537329 and CMT clinicopathological characteristics or survival.

The SNPs assessed herein are synonymous and, thus, do not involve amino acid changes; however, they may influence HER2 gene expression. Although neglected for several years, synonymous genetic variants are nowadays recognized as important players in the pathogenesis, progression, and treatment efficacy of cancer [46,47]. Synonymous SNPs do not alter the protein primary structure (and were, therefore, once termed “silent”) but may affect the conformation and function of the encoded protein. Synonymous SNP may influence mRNA structure, stability and folding, translation kinetics, and access to regulatory factors (e.g., miRNA and RNA-biding proteins), driving changes in protein conformation, phosphorylation, function, and localization [46,47].

Additional, comprehensive studies, enrolling a larger number of subjects (thus allowing adjustments of the genetic profile analysis to different canine breeds), including haplotype studies, assessment of immunohistochemical HER2 expression in mammary neoplastic specimens, and HER2 gene and/or protein expression quantification will be useful to corroborate our data and to explain the influence of the HER2 genetic profile on the pathogenesis, progression, and outcome of CMT.

## 5. Conclusions

This study demonstrated that genetic variation in the canine HER2 gene rs24537331 is associated with a decreased likelihood of tumoral necrosis and with a better outcome of the disease, reflected in a longer disease-specific overall survival. These results are in line with those of a previous investigation, in which we demonstrated an association between this SNP and the development of CMT of less aggressive histotypes [36]. On the contrary, SNP rs24537329 does not seem to be a determinant in the pathogenesis and progression of CMT. In conclusion, our investigation suggests that SNP rs24537331 may be regarded as a potential molecular marker of prognosis in CMTs, able to identify a subgroup of animals prone to develop less aggressive forms of CMT. Our results emphasize the importance of the animal’s genetic background assessment in the toolbox used by clinicians and oncologists in CMT management. They may also help to explain the diversity of results published to date, many of them contradictory, regarding HER2 expression and its association with clinicopathological features of CMT. However, it is important to recall that the diagnostic, predictive, and prognostic value of individual SNPs can be fragile when considered independently. Our investigation will profit from complementary studies integrating additional SNPs, which could contribute to reinforcing its relevance. An integrated approach considering the animal’s genetic background articulated with histological features, tissular and serum HER2 expression, as well as with molecular characteristics of the neoplasms (namely gene and/or protein expression quantification) will allow the development of management strategies tailored to the specificities of the animal and of the tumoral lesion. Indeed, veterinary oncology will certainly benefit from an association between genetic and molecular tests and clinical, image, and histological examinations, similar to human oncology.

## Figures and Tables

**Figure 1 animals-13-01384-f001:**
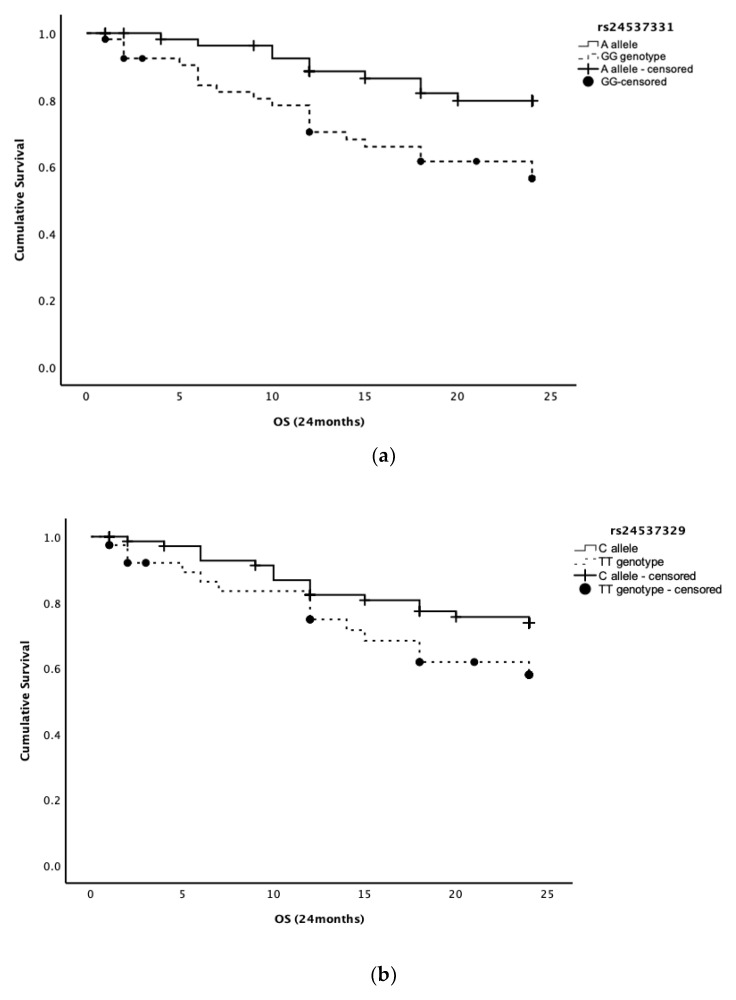
(**a**) Kaplan–Meier plots of the estimate of disease-specific overall survival for variant allele carriers (A allele; n = 60) and wild-type genotype (GG genotype; n = 55) for SNP rs24537331 in HER2 canine gene (*p* = 0.009) in a total population of 115 female dogs with at least one malignant mammary tumor. Censoring is indicated by the + symbol. (**b**) Kaplan–Meier plots of the estimate of disease-specific overall survival for variant allele carriers (C allele; n = 76) and wild-type genotype (TT genotype; n = 39) for SNP rs24537329 in HER2 canine gene (*p* = 0.105) in a total population of 115 female dogs with at least one malignant mammary tumor. Censoring is indicated by the + symbol.

**Table 1 animals-13-01384-t001:** Genotypic and allele frequencies of the SNPs rs24537331 and rs24537329 in the canine HER2 gene.

SNP/Genotypes		Frequency (N)	Percentage (%)
rs24537331	AA	29	14.1
	GG	97	47.3
	GA	79	38.5
	Total	205	100.0
rs24537329	CC	57	27.8
	TT	62	30.2
	TC	86	42.0
	Total	205	100.0

**Table 2 animals-13-01384-t002:** Association between SNP rs24537331 (GG/A allele carriers) and SNP rs24537329 (TT/C allele carriers) in HER2 gene with clinicopathological variables of canine mammary tumors.

	Genotype SNP rs24537331			Genotype SNP rs24537329		
Independent variables	Allele A (n/%)	GG (n/%)	*p*	Allele C (n/%)	TT (n/%)	*p*
Age						
≤10	64 (52.9)	57 (47.1)		82 (67.8)	39 (32.2)	
>10	39 (50.0)	39 (50.0)	0.690	56 (71.8)	22 (28.2)	0.548
Breed (FCI)						
Purebred	55 (49.1)	57 (50.9)		73 (65.2)	39 (34.8)	
Mongrel	53 (57.0)	40 (43.0)	0.260	70 (75.3)	23 (24.7)	0.117
Number of tumors						
Single	41 (57.7)	30 (42.3)		52 (73.2)	19 (26.8)	
Multiple	67 (50.0)	67 (50.0)	0.291	91 (67.9)	43 (32.1)	0.429
Tumor size						
≤3 cm	56 (49.6)	57 (50.4)		77 (68.1)	36 (31.9)	
>3 cm	43 (54.4)	36 (45.6)	0.506	56 (70.9)	23 (29.1)	0.685
Tumor size *						
≤3 cm	30 (47.6)	33 (52.4)		0 (63.5)	23 (36.5)	
>3 cm	36 (55.4)	29 (44.6)	0.379	45 (69.2)	20 (30.8)	0.492
Biological behavior						
Benign	39 (53.4)	34 (46.6)		55 (75.3)	18 (24.7)	
Malignant	69 (52.3)	63 (47.7)	0.874	88 (66.7)	44 (33.3)	0.195
Mode of growth pattern *						
Expansive	28 (54.9)	23 (45.1)		34 (66.7)	17 (33.3)	
Infiltrative	22 (47.8)	24 (52.2)		31 (67.4)	15 (32.6)	
Invasive	19 (54.3)	16 (45.7)	0.755	23 (65.7)	12 (34.3)	0.988
Tumoral necrosis *						
No	45 (60.0)	30 (40.0)		53 (70.7)	22 (29.3)	
Yes	19 (37.3)	32 (62.7)	0.012	30 (58.8)	21 (41.2)	0.169
NHG parameters scores *						
Tubule formation						
Score 1	15 (62.5)	9 (37.5)		16 (66.7)	8 (33.3)	
Score 2	23 (50.0)	23 (50.0)		32 (69.6)	14 (30.4)	
Score 3	14 (40.0)	21 (60.0)	0.236	20 (57.1)	15 (42.9)	0.498
Nuclear pleomorphism						
Score 1	6 (66.7)	3 (33.3)		7 (77.8)	2 (22.2)	
Score 2	34 (51.5)	32 (48.5)		44 (66.7)	22 (33.3)	
Score 3	12 (40.0)	18 (60.0)	0.324	17 (56.7)	13 (43.3)	0.442
Mitotic index						
Score 1	22 (52.4)	20 (47.6)		27 (64.3)	15 (35.7)	
Score 2	11 (39.3)	17 (60.7)		16 (57.1)	12 (42.9)	
Score 3	19 (54.3)	16 (45.7)	0.443	25 (71.4)	10 (28.6)	0.497
NHG						
Grade I	22 (62.9)	13 (37.1)		26 (74.3)	9 (25.7)	
Grade II	19 (40.4)	28 (59.6)		27 (57.4)	20 (42.6)	
Grade III	11 (47.8)	12 (52.2)	0.131	15 (65.2)	8 (34.8)	0.287
Vet-NPI *						
≤4.0	33 (55.0)	27 (45.0)		42 (70.0)	18 (30.0)	
>4.0	19 (42.2)	26 (57.8)	0.195	26 (57.8)	19 (42.2)	0.194
Vascular invasion *						
No	58 (54.2)	49 (45.8)		74 (69.2)	33 (30.8)	
Yes	8 (36.4)	14 (63.6)	0.127	11 (50.0)	11 (50.0)	0.084
Lymph node metastases *						
No	42 (50.0)	42 (50.0)		54 (64.3)	30 (35.7)	
Yes	19 (61.3)	12 (38.7)	0.282	22 (71.0)	9 (29.0)	0.502

Legend: SNP: single nucleotide polymorphism; NHG: Nottingham grade; vet-NPI: veterinary adapted Nottingham prognostic index. * Only malignant tumors were considered.

## Data Availability

The data presented in this study are available on request from the corresponding author.

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
