# Peer review of "Protective Effect of HER2 Gene Polymorphism rs24537331 in the Outcome of Canine Mammary Tumors"

_animals, 2023, doi:10.3390/ani13081384_

Round 1

Reviewer 1 Report

The paper entitled “Protective effect of HER2 gene polymorphism rs24537331 in the outcome of canine mammary tumours” is well written and provides novel information on the possible role of HER2 gene in mammary tumours progression.

I have some suggestions to improve the readability of the paper:

Please add a table reporting characteristics of the canine population included (breed, age), as well as the histological diagnosis.

Did authors investigate a possible association between HER2 polymorphism and breed?

Table 2: delete (n= number of cases) in the headings. Add a column with the total number of cases per each line. Report the exact p value, instead of NS

In figure 1 legend please report the total number of cases considered. Only malignant tumours  were considered in Kaplan-Meier plots?

Reviewer 2 Report

In this study, the authors have investigated 2 single nucleotide polymorphisms (rs24537331 and rs24537329) of the HER2 gene in 205 female dogs with mammary tumors, of which 73 (36%) had benign mammary tumors, and 132 (64%) had malignant mammary tumors. 115 female dogs with malignant mammary tumors and 2-year follow-up data were included in survival analyses. There were very few associations between the studies SNPs in the canine HER2 gene and the clinico-pathological characteristics of mammary tumors, which have been very well characterized in this study. However, the SNP rs24537331 AA was associated with better cancer-specific survival than the wild-type GG genotype (HR=2.59, p=0.013).

This study is innovative, informative, and rigorous; the manuscript is very well written; the number of studied animals is adequate. Very few associations between genetic variations and survival have been reported so far in canine mammary oncology. For these reasons, I strongly support further publication of this manuscript in Animals, and have listed below a few suggestions for minor modifications.

Minor comments

1.          Page 1, lines 24–36: the abstract could be more precise / informative: number of dogs, number of benign versus malignant mammary tumors, frequency of allelic variants, association between SNP rs24537331 and survival (Hazard Ratio, p value); the sentence lines 32–34 is not really useful. 

2.          Page 2, lines 49–51 (introduction): in this sentence which refers to human breast cancer, reference 11 (Yu et al) may not be the most appropriate, maybe add a review on the roles of HER2 in breast cancer (for instance reference 28).

3.          Page 2, lines 55–62: I think it is important to recall also that HER2 overexpression has not been found in canine mammary tumors according to Abadie et al 2018 (IHC) and Burrai et al 2015 (IHC, qRT-PCR, and mass spectrometry). In IHC, most of the conflicting results in the literature result from protocol differences (various primary antibodies, and concentrations/incubation times): in IHC, HER2 overexpression may be overestimated because IHC protocols are too sensitive, and not validated on appropriate breast cancer controls. A valid protocol should give an IHC score of 0 in the normal mammary gland, for instance. References: Abadie J, Nguyen F, Loussouarn D, Peña L, Gama A, Rieder N, Belousov A, Bemelmans I, Jaillardon L, Ibisch C, Campone M. Canine invasive mammary carcinomas as models of human breast cancer. Part 2: immunophenotypes and prognostic significance. Breast Cancer Res Treat. 2018;167(2):459-468. doi: 10.1007/s10549-017-4542-8. And: Burrai GP, Tanca A, De Miglio MR, Abbondio M, Pisanu S, Polinas M, Pirino S, Mohammed SI, Uzzau S, Addis MF, Antuofermo E. Investigation of HER2 expression in canine mammary tumors by antibody-based, transcriptomic and mass spectrometry analysis: is the dog a suitable animal model for human breast cancer? Tumour Biol. 2015;36(11):9083-91. doi: 10.1007/s13277-015-3661-2.

4.          Page 2, line 59 (introduction): the us of “poor prognosis” is debatable here, because the cited references did not include survival studies; maybe replace by something like “some authors associated HER2 overexpression with aggressive clinico-pathological features”. 

5.          Page 3, line 105 (methods), and throughout the text: by definition, the survival used in this study is cancer-specific survival (time from diagnosis to cancer-related death), not overall survival (time from diagnosis to death from any cause).

6.          Page 3, lines 104–109 (methods): please indicate the duration of follow-up.

7.          Page 4, lines 143–145 (results): was the age at diagnosis significantly different between dogs with benign MTs and dogs with mammary cancers?

8.          Page 5, line 168 (results): is the survival really reported as a mean, or a median?

9.          Pages 4–5, Table 2: can the authors please explain why the pattern of growth is only available in 132 cases, not 205? Why is it reported in 132 tumors on the left and 134 on the right? Why is tumor necrosis only reported for 83 tumors? For the histological grade (only reported for 105 tumors), maybe specify again in a footnote that only mammary carcinomas have been graded. Why is vascular invasion reported for 129 tumors, not the 132 malignant tumors? Can the authors please specify if the 115 draining lymph nodes in Table 2 only refer to draining lymph nodes of malignant tumors? Alternatively, 2 separate tables could be provided, one with the 205 tumors (lines age, number of tumors, tumor size), and another one only for the 132 malignant tumors.

10.      Page 6, Figures 1a–1b: maybe specify again in the legend that only the 1st survival curve demonstrates significant survival differences between groups. 

11.      Page 6, line 190 (discussion): is reference 32 appropriate here? Deals with HER2 overexpression/amplification feline pulmonary carcinomas, not with genetic variation of the canine HER2 gene. 

12.      Were there any associations between the studied SNPs and the breed of dogs? 

13.      Page 7, lines 220–221 (discussion): can the authors please comment on the fact that a quantification of either HER2 gene expression, or HER2 protein expression, would be an interesting complement to the present study?

14.      Page 7, lines 230–231: I think that this sentence is an over-interpretation of the results presented: SNP rs24537331 was found associated only with a decreased probability of tumor necrosis, not with many histopathological features considered to be of good prognosis. 

Typos / Spelling

-       Page 2, line 61 (introduction): “HER2” instead of “HER”.

-       Page 3, line 122 (methods): maybe replace “chromosome 9” by “dog chromosome 9” or “CFA9”.

-       Page 3, lines 138–139 (results): “30.2”, “42.0”, “27.8”: with dots instead of commas. Same in Table 1 and in Table 2. 

-       Page 5, line 172 (results): “HR” (Hazard Ratio) instead of OR (Odds Ratio).

-       Page 7, line 200 (discussion): the formulation “influences the presence of” may be not appropriate; what is sure is that the authors have found a statistical association between SNP rs24537331 and tumor necrosis, not that there is a cause-effect relationship. 

-       Page 8, line 268: reference 5 is incomplete (lacks the name of the book).

-       Page 9, line 313, reference 23: the authors are Zappulli V, Peña L, Rasotto R, Goldschmidt M, Gama A, Scruggs J, Kiupel M. the editor is the Davis-Thomson Foundation.

Reviewer 3 Report

This is an interesting paper on the influence of individual genetic profile on mammary carcinogenesis. The authors have experience in this field, as evidenced by several publications on this topic to which they refer, pointing out that it is the same SNP, whereas the SNPs previously studied are different (line 71 ref 10; lines 190-193 ref 22, etc.).
These publications have in common that almost none of the SNPs used is statistically correlated with the malignancy indices considered in the different publications.
In view of this, the results of the present paper show that the only data with statistical significance are the association between the SNP rs24537331 and the presence of necrosis, which is known to be an index of high malignancy and poor prognosis. In contrast, they instead show a prolongation of survival, noting that "our results suggest that this SNP has a protective effect on CMT outcome and is significantly associated with longer survival." The authors do not provide any explanation or hypothesis to explain this, and this can be accepted, but to say that this aspect can be used as a positive prognostic factor seems excessive. The authors should therefore reflect on the actual applicability of these parameters and, above all, not give overstated concusions. 
Some specific comments are listed below.
INTRODUCTION
The statement referring to ref 10 (line 71) is not correct as it refers to SNPs of CD1 encoding E-cadherin and not HER2. Moreover, the SNPs analyzed are different. Therefore, the claim is not supported by a valid reference.
Lines 73-76: The sentence should be corrected as it is not about "further evaluation ... in CMT" but a general goal in oncology (see previous comment)
MATERIAL AND METHODS
Line 87: Number of tumors: do the authors mean multiple tumors in the same mammary gland chain? If yes, indicate whether they were considered as a single neoplasm from a histological point of view.
RESULTS
Authors should clarify any acronyms used in the table (e.g., line 123 T/C change (Glu-Glu) is GG or T/C in the table?)
DISCUSSION
See general comment
REFERENCES Reference # 10 does not contain data. please correct

Round 2

Reviewer 3 Report

Thank you for the revision which now clarifies (among other) the former confusion between tumor necrosis and prognosis, confirming that tumor necrosis is a negative prognostic factor. One additional issue requires an explanation, however: former reference #10 which referred to a paper by your group and which you listed in the original version as "in print" had been published already in 2019; but now, in the present revision, this reference is not anymore listed; deleting this reference may not be an omission; in fact, this paper published in Vet Comp Oncol https://doi.org/10.1111/vco.12510 seems almost a "double publication" as all is quite identical to the present paper (sample and control group, M&M, ethics committee authorisation, authors etc. except a different genetic variation (E-cadherin) was investigated). Please explain and/or include this, your previous, valuable contribution in the present reference list (which would not be undue auto referencing as most other of your papers in the field are listed).   

There are a few textual corrections to be made which you find listed below:

Editor’s corrections

Line 10: spell out first, then abbreviation in parentheses

Line12: are conflicting.

Line 27: …tumor histotypes.

Line 37: .. the clinical image and histological examinations, when assessing CMT outcome.

Line 46: add “human” to “breast cancer”; the term breast cancer should only be used in context of human disease (references 1, 2 are in fact human medicine refs). 

Line 53:..prognostic challenge.

Line 122: 10e75% ??

Line 255: “insipient” ? better use “fragmentary, incomplete, inconclusive” or the like….

Author Response

The authors would like to thank the reviewers for the attention given to the manuscript as well as for all the suggestions that undoubtedly contributed to raising its quality and readability.

According to the Editor recommendation, some new sentences were added to the “Introduction” section, allowing us to reach approximately the required 4000 words.

Regarding the concerns raised by reviewer #3:

As we already mentioned, former reference #10 resulted from an error in the insertion of the list of bibliographical references. The exclusion of this reference from the present version of the manuscript resulted exclusively from this error. We regret that the correction of this error led the Reviewer to argue that the present study is a double publication, which is a very serious assumption. Moreover, the reviewer also demonstrated concerns about the auto-citation of the previous papers of our group. In the next paragraphs we will try to further explain the progress of our group research, so that no more doubts remain:

Our group pioneered the study of the relationship between genetic polymorphisms and canine mammary tumors (CMT) characteristics, namely risk, clinical and histological features, and prognosis. Our first works on this subject date back to 2008 (ref#8 Dias Pereira et al 2008) and 2009 (ref# 10 Dias Pereira et al 2009) so it is inevitable to cite these papers when approaching the subject.

Over the more recent years, we managed to carefully gather a group of animals with CMT from which we were able to obtain clinical information, biological material (blood), mammary neoplasms and follow-up data. An investigation was outlined on the relationship between the animals’ genetic profile and several clinicopathological characteristics of the neoplastic disease, which was approved by the ethics committee of our institution (Ethics Committee of the Institute of Biomedical Sciences Abel Salazar, University of Porto (ORBEA; P151/2016).

The genetic profile of these animals was analysed, specifically SNP research in several different genes related to the carcinogenesis process, generating a large amount of data.

This material was subsequently analysed, assessing its association with the risk of developing CMT (ref #26, Canadas et al 2018), as well as with the type and histological grade of CMT (ref #12, Canadas-Sousa et al 2019).

The relationship between SNP in specific canine genes and various clinical and histological characteristics of CMT was also assessed, as well as with disease progression (survival data), generating several papers, namely ref #11 (Canadas et al 2018) and other references that are not listed in the present manuscript because we do not consider them as directly related to the subject, e.g.: Canadas-Sousa et al 2021 (prolactin gene), Canadas et al 2019 (E-cadherin gene), Canadas-Sousa et al 2019 (ER gene).

Inevitably, in the case of data obtained from the same group of animals and using the same genetic profile analysis techniques, the description of the group and methodology is very similar between our publications. This is very much the same that happens with immunohistochemical studies performed in a certain cohort, in which the expression of different non-related immunohistochemical markers is assessed.

We hope these explanations have resolved the reviewer concerns and dispelled all his doubts.